# High performance data integration for large-scale analyses of incomplete *Omic* profiles using Batch-Effect Reduction Trees (BERT)

Yannis Schumann [1,6] ✉, Simon Schlumbohm[2,6], Julia E. Neumann [3,4,7] ✉ & Philipp Neumann [1,5,7] ✉

Data from high-throughput technologies assessing global patterns of biomolecules (*omic* data), is often afflicted with missing values and with measurement-specific biases (batch-effects), that hinder the quantitative comparison of independently acquired datasets. This work introduces batch-effect reduction trees (BERT), a high-performance method for data integration of incomplete *omic* profiles. We characterize BERT on large-scale data integration tasks with up to 5000 datasets from simulated and experimental data of different quantification techniques and *omic* types (proteomics, transcriptomics, metabolomics) as well as other datatypes e.g., clinical data, emphasizing the broad scope of the algorithm. Compared to the only available method for integration of incomplete *omic* data, HarmonizR, our method (1) retains up to five orders of magnitude more numeric values, (2) leverages multi-core and distributed-memory systems for up to 11 × runtime improvement (3) considers covariates and reference measurements to account for severely imbalanced or sparsely distributed conditions (up to 2 × improvement of average-silhouette-width).

High-dimensional *omic* data refers to high-dimensional, untargeted profiles of an entire distinct biomolecule modality (e.g., genomic data from DNA; transcriptomic data from RNA, or proteomic data from proteins). This *omic* data is frequently used in the biomedical domain for various purposes, such as to characterize molecular changes between healthy and cancerous tissue[1–3]. Restricted data availability, limitations in the employed wet lab pipeline or measurement technique and even financial concerns often lead to small cohorts per study, which do not exhibit the statistical power required for downstream analyses on their own. Therefore, researchers employ data integration to combine their study with other data, which can however introduce (or amplify) data incompleteness, which is commonly presented by *omic* profiles[4–7].

Individual datasets (also referred to as *batches* in the following) usually exhibit technical biases, so-called batch effects, which necessitate the application of data integration algorithms when combining measurements from multiple data sources[8]. While batch-effect correction is well established for some fields (e.g., DNA-methylomics[9,10]), the complexity and experimental variance of recent *omic* technologies (e.g., proteomics[8,11], metabolomics[12,13]) make the development of novel batch-effect reduction methods both necessary and challenging[14,15].

The most prevalent contemporary challenges for large-scale data integration consist in the computational efficiency of the respective batch-effect correction methods[15,16] and in the incompleteness of *omic* data[17,18]. A large number of imputation methods exist for different *omic* types[19–21], but it was repeatedly elucidated that the typical unawareness

[1]Deutsches Elektronen-Synchrotron DESY, Hamburg, Germany. [2]Chair for High Performance Computing, Helmut-Schmidt-University Hamburg, Hamburg, Germany. [3]Center for Molecular Neurobiology Hamburg (ZMNH), University Medical Center Hamburg-Eppendorf (UKE), Hamburg, Germany. [4]Institute of Neuropathology, University Medical Center Hamburg-Eppendorf (UKE), Hamburg, Germany. [5]High Performance Computing & Data Science, University of Hamburg, Hamburg, Germany. [6]These authors contributed equally: Yannis Schumann, Simon Schlumbohm. [7]These authors jointly supervised this work: Julia E. Neumann and Philipp Neumann ✉e-mail: yannis.schumann@desy.de; ju.neumann@uke.de; philipp.neumann@desy.de

of different missing value mechanisms in *omic* data hampers the effectiveness of these methods[22,23]. To this end, researchers presented the imputation-free HarmonizR framework[24], which employs matrix dissection to identify sub-tasks (sub-matrices) that are suitable for *embarrassingly parallel* data integration using the established ComBat[25] and limma[26] methods. In order to construct suitable sub-matrices for some features, HarmonizR introduces additional data loss to the data (default behavior in the latest version, referred to as *unique removal* (UR) in the following). Recent developments of HarmonizR further allow to consider groups of batches as a joint pseudo-batch (blocking approach) in order to improve runtime[27,28]. To the best of our knowledge, HarmonizR is the only data integration tool to date that allows the consideration of arbitrarily incomplete *omic* data and is hence used for benchmarking our method in this work. Of note, HarmonizR has been applied to proteome data, where it was demonstrated to outperform batch-effect correction with internal reference samples[29] and imputation.

Typical data integration tasks concern datasets with variable distribution of covariates between batches, unknown covariate levels for some cases and even unique covariate levels that exclusively occur in one batch. It has been extensively demonstrated that such design imbalance among batches may hamper batch-effect correction and subsequent biological analyses[30–32]. HarmonizR does not provide methods to address such an imbalance yet. The COCONUT framework[33] introduced user-defined references to ComBat, but it cannot handle arbitrary sets of missing values in a dataset.

In this work, we develop an efficient data integration method for incomplete *omic* data that allows for the consideration of design imbalances by means of covariates and user-defined references. To this end, we introduce batch-effect reduction trees (BERT), which is a high-performance, tree-based data integration framework that leverages existing, well-established methods (i.e., ComBat and limma) and combines hierarchical methods with the HarmonizR approach to provide robust batch-effect correction of incomplete *omic* data, cf. Fig. 1. We demonstrate the robustness, efficiency and scalability of the approach on experimental and simulated data.

## Results

### Algorithm and Implementation
For data integration, BERT relies on the algorithms ComBat and limma, which both mandate that each batch exhibits at least two numerical values per feature. BERT relaxes this condition by additionally allowing features to be missing completely per batch. The fulfillment of either condition is ensured by appropriate pre-processing, in which BERT removes singular numerical values from individual batches (typically ≪ 1% of the available numerical values cf. Section Integration of *Omic* Data and Section Discussion).

BERT decomposes any data integration task into a binary tree of batch-effect correction steps. That is, pairs of batches are selected at each tree level and are corrected for their respective batch effects, which ultimately yields one, fully integrated dataset, cf. Fig. 2A (left panel). In each pairwise correction step, BERT applies ComBat or the linear model from limma to features with sufficient numerical data (i.e., at least two numerical values). All other features, for which the numerical values exclusively originate from either one or the other input batch (due to pre-processing, the other batch contains only missing values), are propagated to the next tree level without further changes, cf. Fig. 2A (right panel).

The data flow of BERT is visualized in Fig. 2B. BERT computes quality control metrics of the provided input data. The binary tree described above is then decomposed into independent sub-trees, which are processed independently by a user-defined number $P$ of BERT processes, yielding respective intermediate batches. These are then processed repeatedly using an iteratively reduced number of BERT processes (parametrized by user parameter $R$, representing the

reduction factor for the number of processes) until parallelization stops at a user-defined number $S$ of intermediate batches, which are then integrated sequentially. Finally, BERT reports quality control estimates on the integrated data as well as the elapsed execution time. The algorithm returns the output data in the same order and data type as the original user input. Note that $P$, $R$ and $S$ exclusively control the degree of parallelization and do not influence the output quality.

BERT allows users to specify any number of categorical covariates (e.g., biological conditions such as sex, tumor vs. control, ...), which need to be known for every sample, cf. Fig. 2C. The underlying batch-effect correction algorithms, limma and ComBat, can model such conditions by modification of their design matrices (i.e., the linear system describing the design of the experiment mathematically), which allows to distinguish the effect of the respective covariate level from the batch-effect. BERT passes the user-defined covariate levels to ComBat/limma at each tree level, such that batch effects are removed, whereas covariate levels are preserved. Often, covariate levels are known for only a subset of samples. In such cases, BERT allows users to indicate these samples as *references*. As an example, consider a dataset with two batches, each containing two WNT-medulloblastomas (the references) and a collection of various tumor samples with unknown tumor type (the non-references). A custom limma implementation is used to estimate the batch effect amongst these references for each pair of batches. These estimates are subsequently applied to perform a joint batch-effect correction step of all reference and (covariate-unknown) non-reference samples of the respective pair, cf. Fig. 2D.

BERT has been implemented in R[34] and is thoroughly unit-tested on various releases of all major operating systems (Mac, Linux, Windows). A user-friendly library has been published to Bioconductor and GitHub under the GNU General Public License (GPL), version 3.0 and the code has undergone peer-review during publication to Bioconductor. Reasonable parameter choices are provided as defaults, but may be further specified by the user. BERT accepts multiple standard input types (data.frame, S4 SummarizedExperiment[35]) and reports quality estimates for the raw and integrated data by means of the average silhouette width[36] (ASW)

$$ASW = \sum_{i=1}^{N} \frac{b_i - a_i}{\max(a_i, b_i)}, \quad ASW \in [-1, 1] \tag{1}$$

where $N$ denotes the total number of samples and $a_i$, $b_i$ indicate the mean intra-cluster and mean nearest-cluster distances of sample $i$ with respect to biological condition (referred to as ASW label in the future) or batch of origin (ASW Batch). The ASW score was demonstrated to exhibit broad consensus with other metrics (iLSI, kBET, ARI), and is particularly useful for global data characterization[16].

### Simulation studies
To maintain precise control over the considered data, we first characterized BERT using simulated scenarios. For each dataset, a complete data matrix was generated as described in the Online Methods (Section "Methods") and an appropriately sized subset of features was randomly chosen to be completely present or missing per batch. The validity of this MCAR (missing completely at random) scheme was validated in additional experiments (cf. Supplementary Information, Section 3), which confirmed that the observed characteristics of batch-effect correction quality generalize to typical MNAR (missing not at random) schemes caused by e.g., detection thresholds.

In 10 repetitions, datasets with 6000 features, 20 batches with 10 samples each and two simulated biological conditions (also referred to as *label* or *class* in the following) were generated. The ratio of missing values was varied up to 50%, and the number of retained numeric values and the sequential execution time of BERT and HarmonizR (full dissection and blocking of 2 or 4 batches, default UR for optimum HarmonizR results) was compared. BERT was observed to retain all

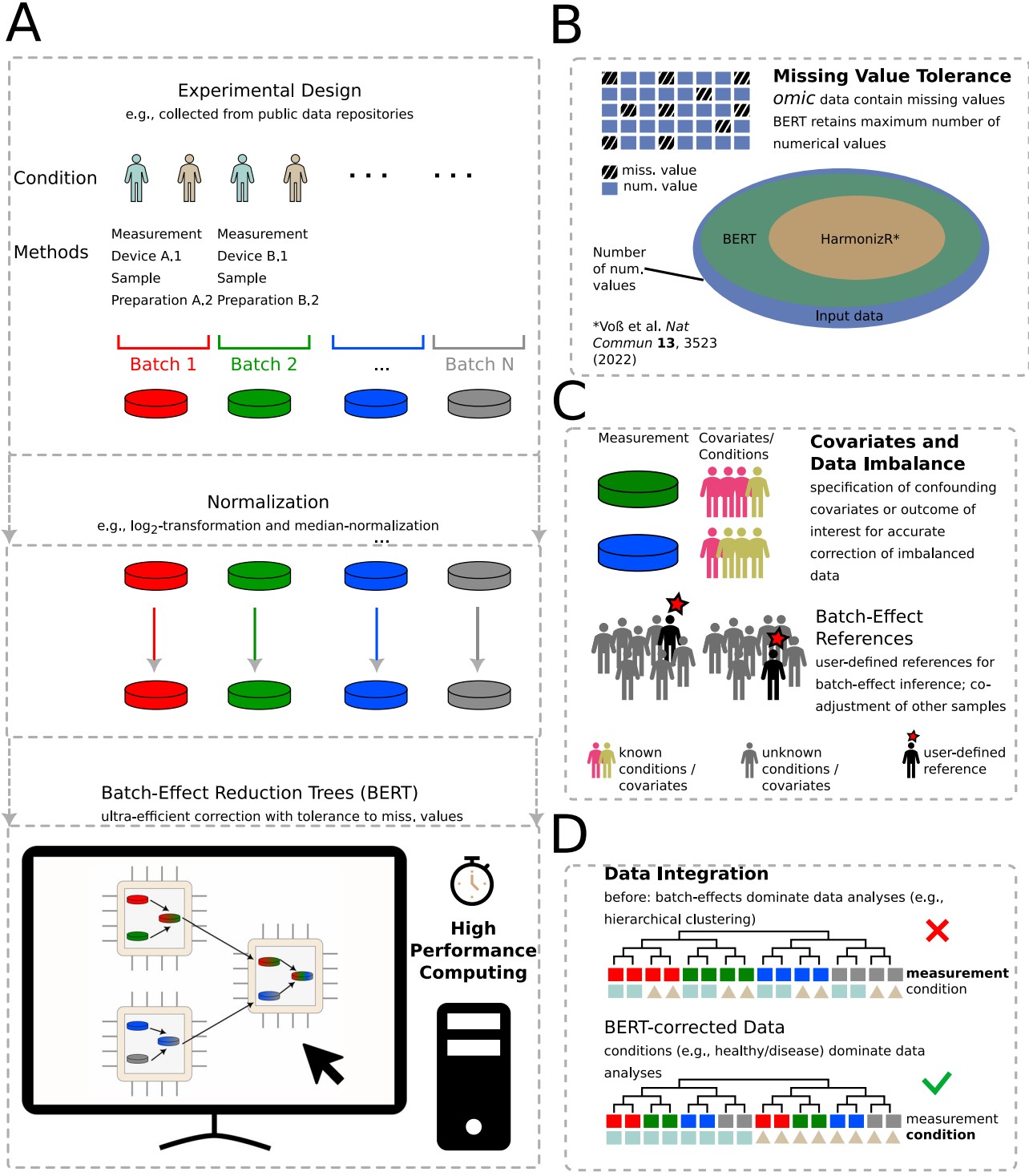

**Fig. 1 | Illustration of the Batch-Effect Reduction Trees (BERT) algorithm.**
**A** Independent measurements are passed to the BERT algorithm, which leverages contemporary multi-core architectures for efficient, hierarchical data integration. **B** Omic data is typically afflicted with missing values. While the state-of-the-art HarmonizR method requires the neglect of large portions of numerical input data, BERT facilitates data integration with minimal loss of data. **C** Uneven distribution of covariate levels (e.g., sex, age certain biological conditions), can negatively affect data integration. BERT allows for the consideration of categorical covariate levels (top, covariates known for all samples) and user-defined references if covariates are not known for the entire dataset (bottom, covariate level unknown for non-starred samples). **D** Concept sketch: Batch effects typically mask biological conditions in the raw data (top). Data integration with BERT alleviates batch effects, making biological signals more prominent for downstream data analyses (e.g., hierarchical clustering).

numeric values, whereas HarmonizR exhibited higher data loss with increasing number of missing values (max. 27% loss for full dissection and 88% data loss for blocking of 4 batches), cf. Fig. 3A.1. On complete data, BERT results were equal to HarmonizR (max. abs. difference of 0.01 and 0.09 for ASW batch and ASW label scores, respectively (limma)). Generally, BERT was faster than HarmonizR for each number of missing values and for each blocking strategy of HarmonizR (full dissection, blocking of 2 or 4 batches). Limma was observed to yield lower runtime than ComBat (e.g., average 13% improvement for BERT). While the execution time of BERT decreased with increasing numbers

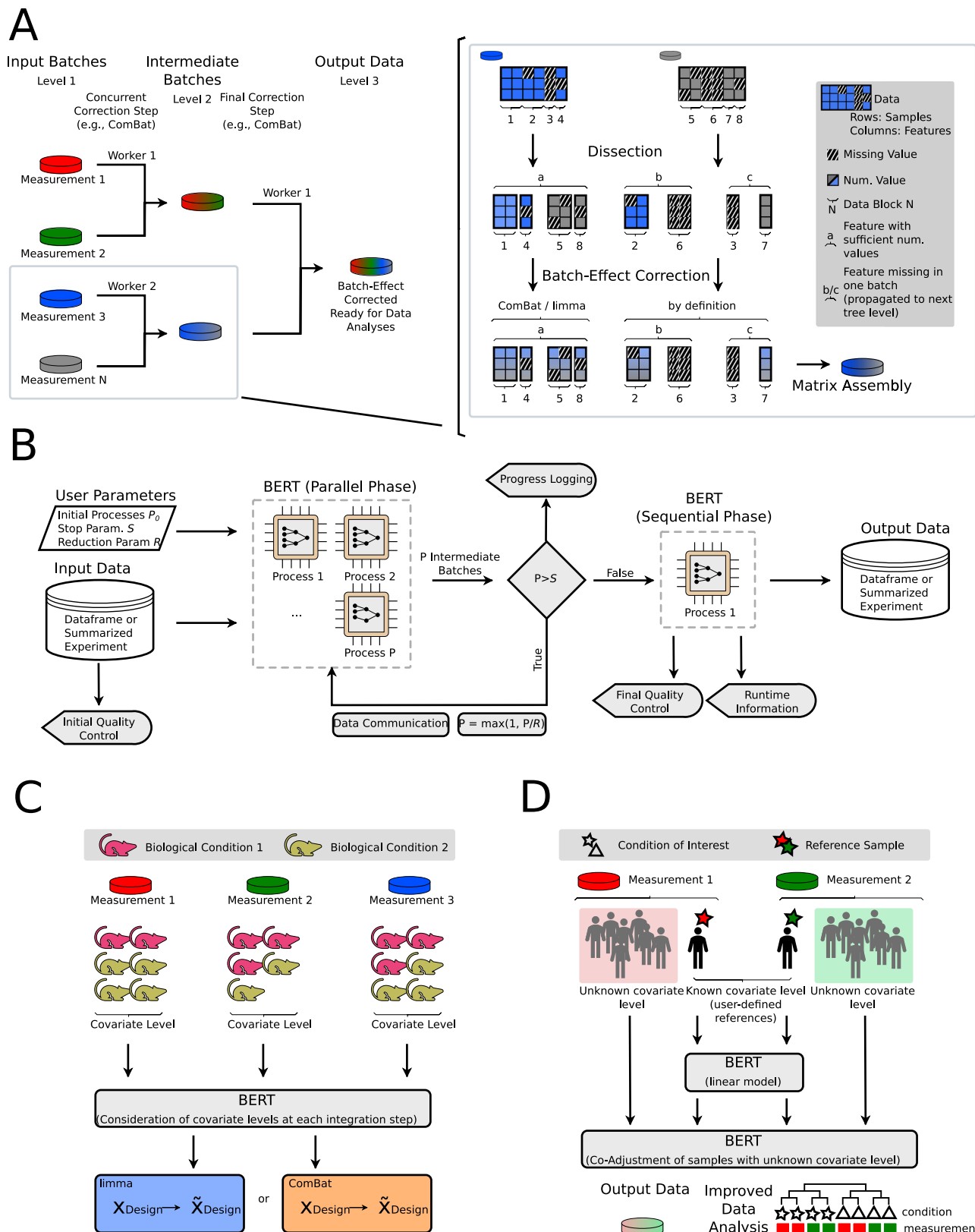

**Fig. 2 | Detailed overview over the BERT algorithm. A** Input batches are processed in a binary tree of pairwise batch-effect correction steps (left) using ComBat or limma for features with sufficient numerical values from the respective input batches, whereas features with fully missing data in either of the batches are propagated to the next tree level without further changes, since batch-effects can not be quantified without another batch to compare (right). **B** BERT first processes the input datasets in a parallel phase, in which sub-trees are integrated on an iteratively decreasing number of independent BERT processes, followed by a sequential phase, in which the remaining intermediate batches are integrated into the final output data. **C** Biological and other known conditions can be specified as categorical covariates and will be considered accordingly by the underlying batch-effect correction algorithms. **D** Users may specify arbitrary samples as references, for which the batch effect is modeled linearly by BERT, followed by the co-integration of all other, non-reference samples.

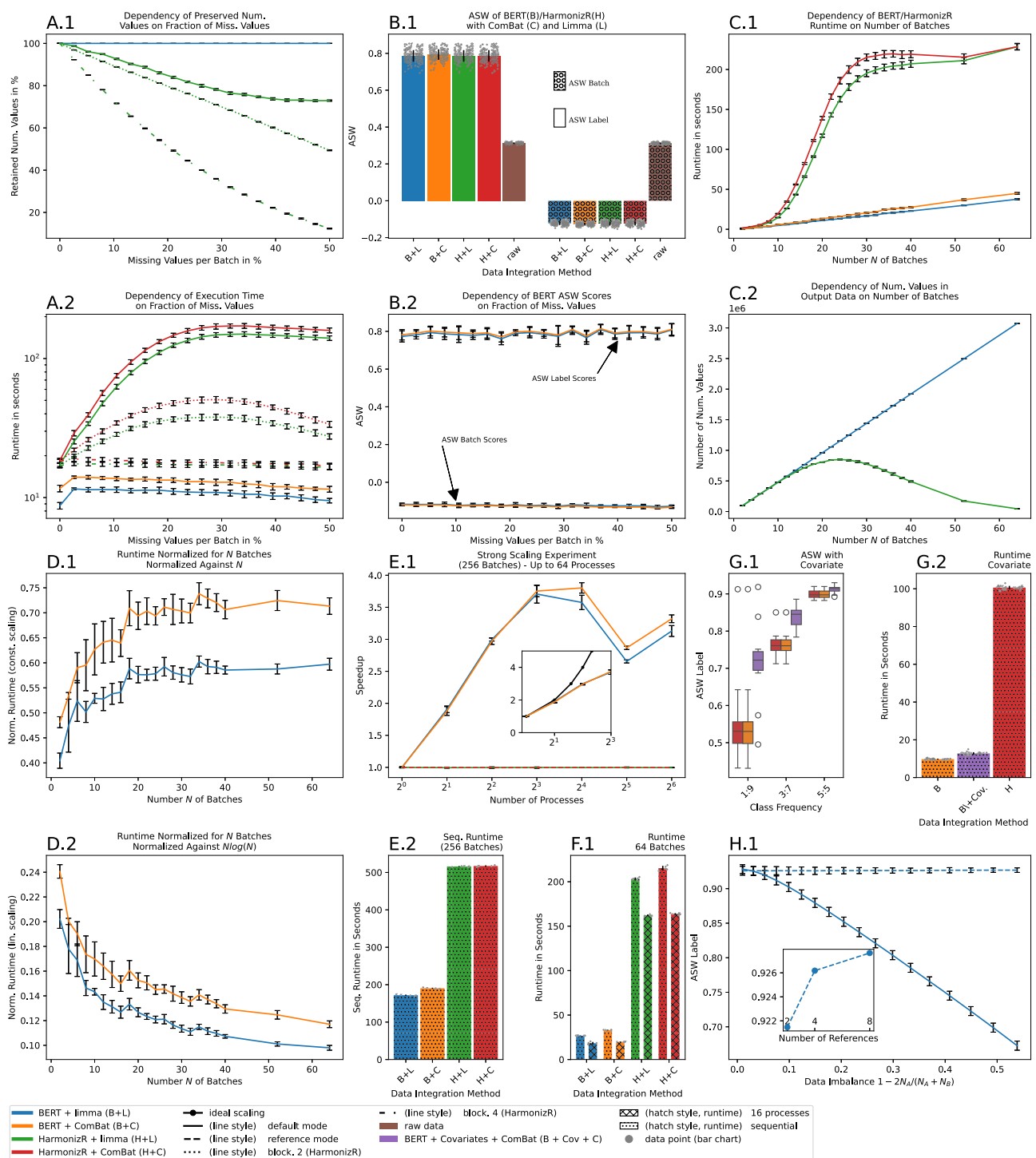

of missing values (e.g., 5 s maximum difference of average runtime for limma), the runtime of HarmonizR increased by up to 8× with the number of missing values per batch (162 s absolute difference), cf. Fig. 3A.2. BERT and HarmonizR both successfully removed the batch-effect and improved the biological signal, as indicated by ASW scores (cf. Fig. 3B.1) as well as by results of an exemplary classification task (cf. Supplementary Information, Section 4). Both metrics indicated approximately constant scores for an arbitrary number of missing values (cf. Fig. 3B.2 for ASW scores) and did not show significant differences between BERT or HarmonizR (cf. Supplementary Information Section 3, Fig. 3A.1 for comparison with MNAR type data).

In a similar experiment, multiple datasets with up to 64 batches and 20% missing values per batch were generated, while all other parameters were left unchanged. The sequential execution time of HarmonizR (full dissection) was observed to increase strongly with the number $N$ of batches (abs. max. difference of 268 s), whereas the runtime of BERT remained much lower, cf. Fig. 3C.1. Note that the absolute number of numeric values retained by HarmonizR decreases for a large number of batches, cf. Fig. 3C.2. Normalizing the execution time of BERT to $N$ (linear scaling) or $N \log N$ (log-linear scaling) revealed, that BERT exhibited linear time complexity with respect to the number of batches (indicated by convergence in Fig. 3D.1 and non-convergence in Fig. 3D.1). Additional experiments and theoretical analyses found that BERT preserves *relative* differences in batch-effect corrected data independent of batch order (e.g., mean ASW label difference of $\mathcal{O}(10^{-17})$ for limma), which additionally indicates high

**Fig. 3 | Results from simulation studies.** Mean (indicated by lines/bars) and standard deviations (error bars) refer to 10 independent repetitions of the experiments. **A** Percentage of retained missing values (**A.1**) and sequential runtime (**A.2**) for data integration of 20 batches (10 samples, 6000 features each) with a variable percentage of missing values using BERT and HarmonizR (full dissection or blocking) with either ComBat or limma. **B** Average silhouette width (ASW) with respect to batch of origin (ASW Batch) or simulated condition (ASW Label) for raw data, or integrated data from BERT (B) and HarmonizR (H) using either ComBat (C) or limma (L) for the experiments in panel (**A**). Panel (**B.1**) shows averaged results, whereas (**B.2**) shows the results for BERT (limma and ComBat) across all missing value scenarios. **C** Sequential runtime (**C.1**) and absolute number of numeric values (**C.2**) using BERT and HarmonizR (full dissection) for data integration tasks with up to 64 batches (10 samples, 6000 features each). **D** Sequential runtime of BERT with ComBat and limma, normalized against the number $N$ of batches (**D.1**) and against $N \log N$ (**D.2**). **E** Speedup of BERT (B) and HarmonizR (H, full dissection) using ComBat (C) or limma (L) for data integration of 256 batches (10 samples, 6000 features each) and between 1 and 64 processes. The inset shows the BERT results on an interval of up to 8 processes as well as ideal scaling. **F.1** Runtime of BERT (B) and HarmonizR (H, full dissection) using ComBat (C) or limma (L) for data integration with 64 batches (10 samples, 6000 features each) and either 1 process (sequential) or 16 processes. **G** ASW scores of naïve BERT with ComBat (B), BERT with conditions as covariates and ComBat (B+Cov.) and HarmonizR (H, full dissection) on data integration of 8 batches (80 samples, 6000 features each) with class distribution of 1:9, 3:7 and 5:5 (panel **G.1**, minority class chosen at random for each batch individually). The center line of the boxplots indicates the median, whiskers contain data within [−1.5, 1.5] interquartile range and outliers are represented by circular markers. Panel (**G.2**) depicts the sequential runtime for the experiments from (**G.1**). **H.1** ASW scores with respect to label for naïve BERT and for BERT using 2 reference samples per batch (1 ref. per simulated condition) for data integration of 2 batches (600 features) with variable class imbalance. The inset depicts the average ASW across all imbalance configurations for a variable number of references. Source data are provided as a Source Data file.

---

robustness to error accumulation (cf. Supplementary Information Section 1 for details). We argue that any constant offset would be irrelevant for the anticipated workflows and downstream tasks of users, including differential expression analysis, classification and others, which exclusively rely on relative differences of expression values between samples. Moreover, users will typically apply domain-specific normalization procedures to each feature after batch-effect correction, such as z-score normalization, which diminishes the introduced offset.

Strong scaling experiments were conducted in 10-fold repetition using 256 batches with 10% missing values (all other parameters as reported before). The BERT parameters $R$, $S$ can affect runtime considerably and were optimized for mSection. 2 *for recommendations on parameter choices*. Using limma and ComBat, BERT achieved a maximum speedup of 3.7× and 3.8×, respectively, cf. Fig. 3E.1. For four processes, which are realistic on commodity office hardware, the respective strong scaling efficiency was 75% and 74%, respectively. No speedup was observed for HarmonizR (full dissection), and the sequential runtime was approximately 2.7× higher than for BERT. For 64 processes, BERT was 9.1× faster than HarmonizR (ComBat). On a smaller data integration task with 64 batches and up to 16 processes, a minor speedup was observed for HarmonizR (1.3× for both ComBat and limma), cf. Fig. 3F.1. However, the optimal HarmonizR configuration with limma (ComBat) was 11.1× (10.3×) slower than the respective BERT counterpart (here: $R = 8$, $S = 16$). Of note, larger problems can yield even higher BERT speedups - e.g., a speedup of 8× (5025 s sequentially and 618 s in parallel) was observed using BERT with ComBat on 128 MPI-processes on 2 compute nodes of the supercomputer HSUper (5000 batches with 5000 features, $R = 2$, $S = 8$, BERT v0.99.5).

In order to analyze the effects of covariate-based adjustment, datasets with 8 batches of 80 samples each were generated (two simulated biol. conditions). We considered class imbalance scenarios of 1:9 (strong imbalance), 3:7 (moderate imbalance) and 1:1 (balanced data), where the minority class was chosen for each batch independently. Using ComBat, severely reduced ASW scores (with respect to label) were observed for both HarmonizR and naïve BERT. These scores were improved strongly by passing the conditions as categorical covariates to BERT (cf. Fig. 3G.1), which had only minor effects on sequential runtime (approx. runtime difference of 3.4 s, cf. Fig. 3G.2). In a similar experiment, variably unbalanced datasets with 2 batches, 600 features, 2 simulated conditions and 10% missing values were generated. Class imbalance was introduced by removing a variable amount of samples of one condition from the first batch, and vice versa for the second batch. It can be seen from Fig. 3H.1, that batch-effect correction quality (ASW label) of naïve BERT with limma degrades rapidly with increasing data imbalance, whereas it remains constant using the reference-based approach. Generally, larger

numbers of references performed favorable but the effect was very small ($\mathcal{O}(10^{-3})$).

For validation, we additionally considered simulation scenarios by other authors[37], designed to be particularly challenging for batch-effect correction (100 datasets with an average of 59% missing values, cf. Section Methods). Using limma, BERT yielded an average improvement of $0.13 \pm 0.02$ and $0.42 \pm 0.05$ for ASW Label and ASW Batch, respectively, as well as significant improvements of $F_1$-scores in differential-expression analyses (paired $t$ test $p < 10^{-50}$, Cohen's $d > 4.0$).

## Integration of *Omic* Data

In order to test our algorithm on real experimental data, we selected four datasets from representative quantification techniques (cf. Sec. Methods) in the field of bottom-up proteomics (in the following referred to by the name of the respective corresponding authors, *Krug*[38], *Petralia*[39], *Nusinow*[40], *Voss*[24]) as well as an exemplary dataset from transcriptomic profiling with micro-arrays (composed from multiple studies of ovarian carcinoma and hence denoted *Ovarian* in the following, cf. Section "Methods"). The datasets are composed of up to 42 batches as well as up to 3502365 missing values and are summarized in Fig. 4A.1. For completeness, we report HarmonizR results with and without UR (full dissection).

On the large proteomic datasets (*Krug*, *Petralia*, *Nusinow*), HarmonizR (deactivated UR) introduced high data loss (11%, 29%, 34% of numerical input values, respectively), whereas HarmonizR (default UR) retained more input values (loss of 2%, 6%, 10% numerical input values, respectively), cf. Fig. 4B.1. BERT data loss occurred only during data pre-processing and was orders of magnitude lower than both HarmonizR variants (max. 1% absolute loss across all datasets; up to five orders improvement of relative amount of retained values compared to HarmonizR). Both BERT and HarmonizR removed a small number (1%) of numerical values during pre-processing of the *Voss* dataset, while no data loss occurred for the Ovarian dataset. The sequential execution time of BERT and HarmonizR varied strongly between the datasets, Fig. 4B.2. Generally, data integration with ComBat was slower than with limma (average 6% difference). Using ComBat, deactivation of UR improved HarmonizR runtime on average by 22% (limma: 20%), but BERT was the fastest algorithm on all datasets (average 71% and 72% improvement over default HarmonizR, respectively). Scaling experiments (10 repetitions) with the *Nusinow* dataset on up to 10 processes showed that all algorithms benefit from parallelization, but BERT was always faster than the HarmonizR variants (with and without UR), cf. Fig. 4B.3.

Exemplarily, a t-SNE for the raw and integrated *Krug* data was computed using pairwise Euclidean distances, respectively (cf.[41] for a discussion of t-SNE limitations, esp. for single-cell data). The samples were strongly grouped by the batch of origin in the t-SNE of the raw data (cf. Fig. 4C.1), whereas they primarily grouped by their biological

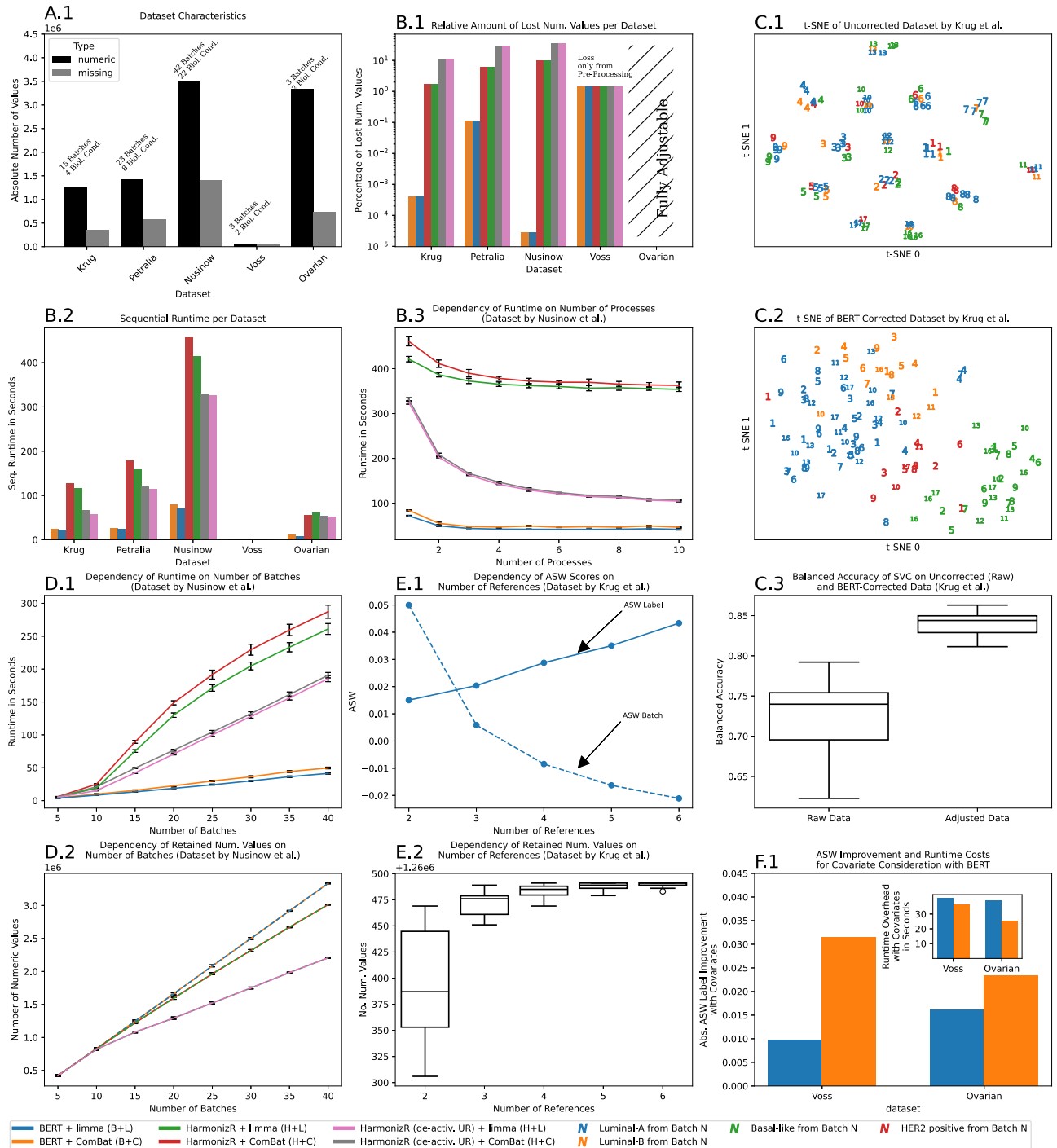

**Fig. 4 | Results from characterization on proteomic data (Krug[38], Petralia[39], Nusinow[40], Voss[24]) and transcriptomic data with micro-arrays (Ovarian, cf. Sec. Methods for references).** If applicable, mean (indicated by lines/bars) and standard deviations (error bars) refer to 10 independent repetitions. For boxplots, the center line indicates the median, whiskers contain data within [−1.5, 1.5] interquartile range and outliers are represented by circular markers. Bar charts correspond to individual data points. **A.1** Absolute number of numerical and missing values per dataset. Textual annotation further describes the respective number of batches and the number of biological conditions. **B** Relative amount of lost numerical values (B.1, log. *y*-axis), sequential execution time (**B.2**) per dataset and the dependency of absolute runtime on the number of processes for BERT and HarmonizR (**B.3**, default and with deactivated UR, ComBat and limma). **C** t-SNE based on pairwise Euclidean distances for the raw (**C.1**) and BERT-corrected Krug-data (**C.2**). Panel C.3 shows the balanced accuracy of a support-vector-classifier on both raw and corrected data in 10-fold leave-one-batch-out cross-validation. **D** Dependency of the sequential runtime (**D.1**) and number of retained numerical values (**D.2**) of HarmonizR (default, deactivated UR) and BERT (Nusinow data, randomly sub-sampled batches) for both ComBat and limma. **E** Dependency of ASW scores (**E.1**) and number of retained numerical values (**E.2**) of BERT on the number of user-defined references (Krug data, random integration of one batch to all other available batches). **F.1** Improvement of BERT ASW scores of (ComBat, limma) when using biological conditions as covariates (Voss, Ovarian). The inset shows the respective increase of sequential runtime in seconds. Source data are provided as a Source Data file.

**Table 1 | Average silhouette width (ASW) of raw, BERT-corrected and HarmonizR-corrected (deactivated and default UR) proteomic data (Krug, Petralia, Nusinow, Voss) and transcriptomic data with micro-arrays (Ovarian)**

| | | ASWBatch | | | | ASWLabel | | | |
|---|---|---|---|---|---|---|---|---|---|
| | | BERT | HarmonizR (default) | HarmonizR (no UR) | raw | BERT | HarmonizR (default) | HarmonizR (no UR) | raw |
| | ComBat | −0.1194 | −0.1208 | −0.1067 | | 0.0486 | 0.0506 | 0.0514 | |
| Krug | limma | −0.1527 | −0.1542 | −0.1401 | 0.3173 | 0.0452 | 0.046 | 0.047 | 0.0065 |
| | ComBat | −0.0968 | −0.1095 | −0.0802 | | −0.0425 | −0.0426 | −0.0356 | |
| Nusinow | limma | −0.1372 | −0.142 | −0.1085 | 0.0669 | −0.0414 | −0.042 | −0.034 | −0.0506 |
| | ComBat | −0.0314 | −0.0198 | −0.0198 | | 0.1762 | 0.1773 | 0.1773 | |
| Ovarian | limma | −0.0211 | −0.0211 | −0.0211 | 0.5055 | 0.1989 | 0.1989 | 0.1989 | 0.1102 |
| | ComBat | −0.1049 | −0.1082 | −0.0862 | | 0.0826 | 0.0832 | 0.0883 | |
| Petralia | limma | −0.1612 | −0.1685 | −0.1457 | 0.2541 | 0.0748 | 0.0775 | 0.083 | 0.0343 |
| | ComBat | −0.1494 | −0.1423 | −0.1423 | | 0.1513 | 0.1536 | 0.1536 | |
| Voss | limma | −0.1462 | −0.1462 | −0.1462 | 0.4308 | 0.1412 | 0.1412 | 0.1412 | 0.0116 |

ASWs are computed with respect to biological condition (ASW Label) or batch of origin (ASW Batch). Source data are provided as a Source Data file.

label (PAM50 breast cancer type[42]) on the BERT-corrected data (cf. Fig. 4C.2, ComBat). A support-vector-classifier (cf. Section "Methods") showed significantly higher balanced accuracy in ten repetitions of leave-one-batch-out cross-validation on the BERT-corrected data compared to the raw data (average of 0.84 and 0.74, respectively; $p < 2 \cdot 10^{-11}$, one-tailed t-test, Cohen's $d \approx 331$), cf. Fig. 4C.3.

Between five and 40 batches of the *Nusinow* dataset were randomly selected for data integration (10 repetitions). The sequential execution time of BERT increased linearly and was considerably lower than for HarmonizR (max. 283 s and 186 s difference, respectively, for default and deactivated UR, ComBat), cf. Fig. 4D.1. BERT retained the largest number of numeric input values, while data loss of HarmonizR increased with the number of batches (up to 10% and 34% loss of HarmonizR compared to BERT's output values using default or deactivated UR, respectively), cf. Fig. 4D.2.

In 50 repetitions, between one and six references were randomly selected to integrate one randomly chosen batch from the *Krug* data to the remaining batches. The ASW scores with respect to both batch and biological label improved consistently with the number of references, cf. Fig. 4E.1. Simultaneously, the number of retained numeric values improved by $\mathcal{O}(100)$, cf. Fig. 4E.2.

Only minor differences ($(O)(10^{-3})$) were observed between the ASW scores of BERT and HarmonizR (deactivated and default UR) on all considered datasets and all algorithms improved data integration quality compared to the raw data, cf. Table 1. Again, ComBat was observed to yield improved batch-effect correction quality over limma. Passing the biological conditions of each sample as a covariate to BERT improved the respective ASW scores on the integrated data (max. improvement of 21% (8%) for ComBat (limma) on datasets *Voss*, *Ovarian*). The consideration of covariates increased the total execution time of BERT at maximum by 36% (41%) on both datasets, Fig. 4F.1.

In addition, three multi-batch metabolomic studies[43] of *Arabidopsis Thaliana* were selected from public data repositories, cf. Sec. Methods. Each dataset represented a different quantification technique (LC-MS, GC-MS, GC-ToF-MS) and different target compounds (classical, volatile and polar metabolites). The raw data grouped primarily by batch in a t-SNE, whereas no batch-grouping was observed in the integrated data (biological labels were unavailable), cf. Fig. 5A–C. ASW improvements with respect to the batch of origin confirmed this quantitatively, and limma performed better than ComBat for BERT and HarmonizR (all quantification techniques). Data loss of BERT was approx. $4 \times - 6 \times$ lower than for HarmonizR (default UR), which in turn had $5 \times - 9 \times$ lower data loss than HarmonizR with deactivated UR, cf. Fig. 5E.1. HarmonizR (default and deactivated UR) had considerably higher sequential runtime than BERT on all datasets, cf. Fig. 5F.1.

Finally, as an example application to non-*omic* data, we evaluated BERT on the ADHD200 cohort[44], i.e., on phenotypic clinical data from social sciences (cf. Supplementary Information Section 5 and Section "Methods"). After BERT, cumulative density estimates of inattentiveness and hyper/impulsiveness were more similar between medical centers (including lower standard deviations), and the ASW with respect to batch improved considerably.

## Discussion

We introduced Batch-Effect Reduction Trees (BERT) – a high-performance method for large-scale data integration of incomplete *omic* data. BERT decomposes the data integration task into a binary tree of pairwise batch-effect correction steps, for which the method employs the state-of-the-art algorithms ComBat and limma, and leverages the multi-core and distributed-memory architecture of contemporary desktop computers and supercomputers. BERT is publicly available as a library for the R programming language via Bioconductor and GitHub under GPL-3.

Simulation studies and data integration tasks on representative single-*omic* datasets from bottom-up proteomics, micro-array-based transcriptomics, and metabolomics highlighted that BERT typically strongly improved the relative amount of retained numerical values compared to the state-of-the-art HarmonizR algorithm (with and without UR). This improvement was even more prevalent when HarmonizR grouped multiple batches together (blocking).

BERT's batch-effect correction quality (ASW) was independent of the number of missing values and equivalent to HarmonizR. Moreover, BERT improved downstream data-analysis tasks such as the biological validity of results from dimensionality reduction (t-SNE) and the performance of classification (SVC). BERT offers new features for the integration of data with unequal class distribution. Users can specify categorical covariates (e.g., outcome of interest or confounders) that are known for each sample or can indicate reference samples with known covariate levels that are used to model the batch-effect and subsequently co-correct all other (unspecified) samples. Covariates were found to improve the ASW scores on representative omic data considerably (both experimental and simulated), while introducing negligible runtime overhead. Similarly, the use of references was demonstrated and larger numbers of references performed favorable with respect to the number of retained numeric values and batch-effect correction quality. It can be concluded that, in scenarios with strong class imbalance, users should aim to specify applicable covariate levels (if known for all samples) or select all samples with known conditions as references to foster optimal batch-effect estimation.

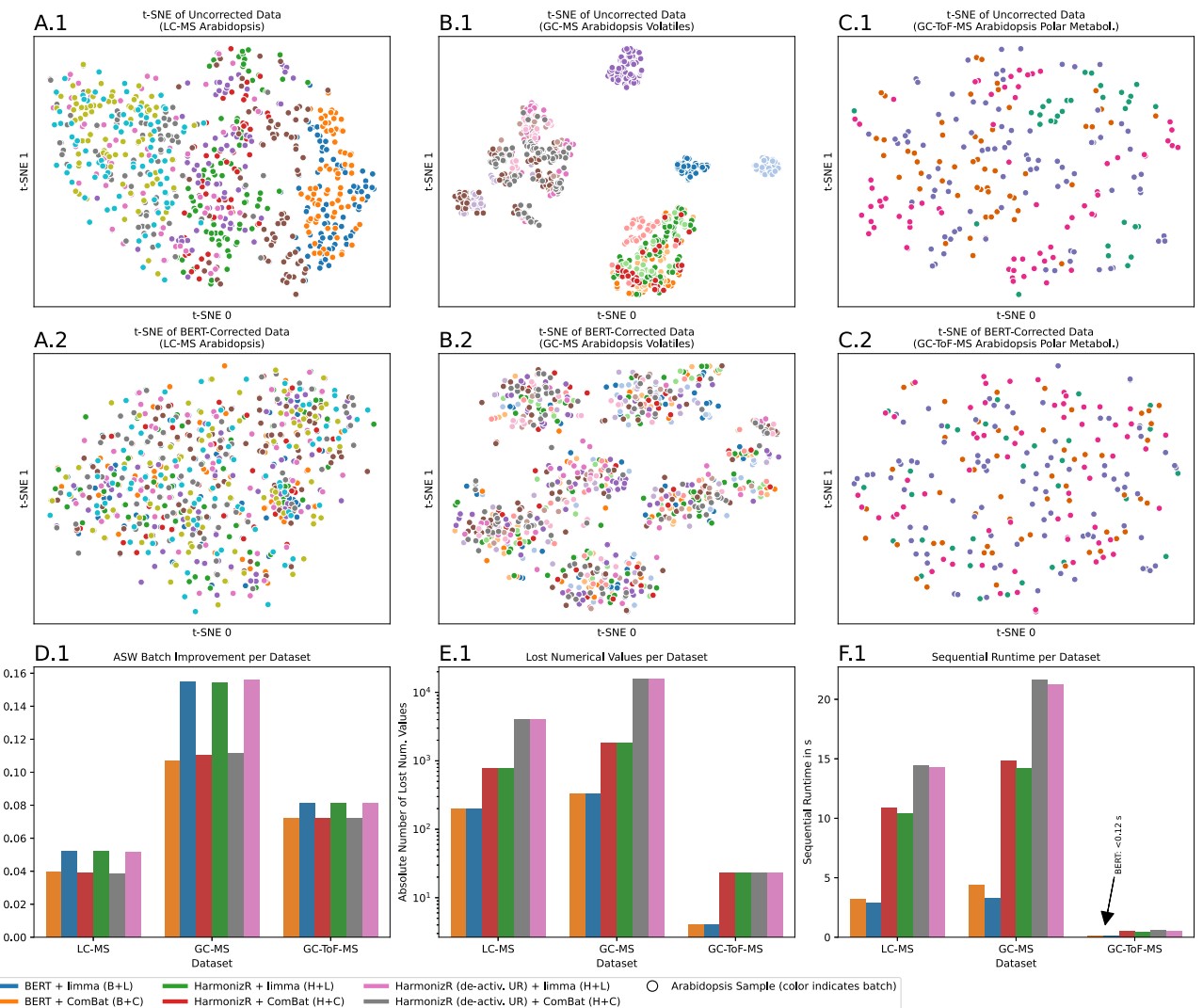

**Fig. 5 | Characterization of BERT on metabolomic data of *Arabidopsis Thaliana*.** Barcharts correspond to individual data points. **A** t-SNE of raw (**A.1**) and BERT-corrected (**A.2**) LC-MS data based on Euclidean distances (marker color indicates batch of origin, biological labels were not available). **B** t-SNE of raw (**B.1**) and BERT-corrected (**B.2**) GC-MS data of volatile metabolites based on Euclidean distances. **C** t-SNE of raw (**C.1**) and BERT-corrected (**C.2**) GC-ToF-MS data of polar metabolites based on Euclidean distances. **D.1** Absolute improvement of ASW scores with respect to batch of origin for BERT and HarmonizR (default and deactivated UR) with limma or ComBat. **E.1** Absolute number of lost numerical values of BERT and HarmonizR (default and deactivated UR) per dataset. **F.1** Sequential runtime of BERT and HarmonizR (default and deactivated UR) per dataset (ComBat and limma). Source data are provided as a Source Data file.

Compared to HarmonizR, BERT offered a typical 4× to 10× improvement for the sequential runtime on the considered omic and simulated data. The algorithm exhibited good strong scaling for realistic numbers of processes on commodity hardware. BERT used MPI to perform data integration of up to 5000 batches on a distributed-memory, multi-core setup on the supercomputer HSUper, which is not feasible with other contemporary algorithms (e.g., HarmonizR, or ComBat/limma alone). The algorithm was observed to exhibit linear time complexity with respect to the number of batches, which highlights its peculiar suitability for large-scale data integration tasks. Moreover, this observation suggested that the execution time of an individual batch-effect correction step may be approximately constant (e.g., limited by overheads of ComBat/limma – cf. Section 6 of Supplementary Information for a detailed analysis) and that BERT might benefit from further improvements of these underlying algorithms in the future. In contrast, the execution time of HarmonizR increased along with both the number of missing values and the number of batches, most presumably due to the correspondingly growing number of sub-matrices.

BERT equips researchers with a tool to reliably perform data integration of incomplete *omic* data on a large scale in order to improve downstream analyses. The algorithm will hence foster the establishment of large-scale reference datasets (e.g., of patients with specific diseases) and will therefore directly contribute to new biological and bio-medical discoveries (e.g., new biomarkers or therapeutic targets). BERT has been established as a fixed element of the research pipeline at the *Core Facility Mass Spectrometric Proteomics* at the University Medical Center Hamburg-Eppendorf.

The peculiar flexibility of BERT allows to adapt the algorithm to various problems at hand, including small-scale and large-scale problems with or without missing values and even data integration tasks with variable class distributions among batches. With respect to the latter, BERT is the first algorithm to date that combines the consideration of covariates and user-defined reference samples with missing-value tolerant data integration. In the upcoming era of large-scale automated analyses and increasing batch sizes (e.g., the rapid increase in sample size using tandem mass tags[45–47]), BERT will facilitate the maintainability and feasibility of dynamic data

exploration and research pipelines. BERT's computational efficiency and scalability on multi-core and shared-memory systems make the algorithm future-proof even for drastic improvements in experiment throughput.

Of note, this study evaluated BERT on three representative omic types, but the underlying algorithms, ComBat and limma, can also be applied to other data modalities. Further analysis of BERT on phenotypic data of the ADHD200 cohort demonstrated considerable improvement of ASW scores and cross-center distribution similarity (cf. Supplementary Information Section 5). This demonstrates that BERT can even be applied to non-omic data, which highlights the peculiar broad applicability of the algorithm and thus extends its capabilities far beyond this work. Prospectively, BERT will also be extended towards batch-effect correction algorithms for sequencing data (e.g., ComBat-seq[48]), which will increase the applicability of the method even further (e.g., scRNA-seq).

In practice, *omic* datasets comprise a mixture of different missing value mechanisms (MCAR, MAR, MNAR)[7,49]. The presented simulation studies employ a feature-wise MCAR scheme, although additional experiments in the Supplementary Information indicate the validity of the results for other, representative MNAR data (e.g., limits of detection). Yet, although our experiments on *omic* data confirmed the applicability of BERT on real data, further validation on simulated MAR data, as well as on controlled mixture scenarios, represents an interesting target for future work.

Throughout the study, batch-effect correction quality was primarily estimated using the ASW scores. Although the validity of the reported improvement with BERT was confirmed by additional experiments and downstream analyses tasks (e.g., classification on simulated and *omic* data, t-SNE plots, Euclidean distances and differential expression analyses), users should generally consider additional metrics when evaluating BERT for their data.

In summary, this work presented a highly efficient algorithm for large-scale data integration and validated it on simulated and experimental (bottom-up proteomics, transcriptomics, metabolomics) data. Current and future research include investigation and runtime-reduction of the underlying batch-effect correction steps and the extension towards sequencing data (especially scRNA-seq). Prospectively, we envision BERT to be integrated in routine research and diagnostic data analysis workflows.

## Methods
### Simulation data
For our simulation studies, we employed the generalized L/S model of ComBat, where the expression value $y$ of feature $f$ for sample $j$ from batch $i$ is given by

$$y_{ijf} = \alpha_f + \mathbf{X}\beta_f + \gamma_{if} + \delta_{if}\epsilon_{ijf}. \tag{2}$$

Here, $\alpha$ is the vector of overall expression values and $\beta$ is the vector of regression coefficients corresponding to the design matrix $\mathbf{X}$. Moreover, $\gamma$ and $\delta$ denote the additive and multiplicative batch-effects and $\epsilon$ represents a normally distributed error term. In our experiments, $\alpha$, $\beta$, $\gamma$ were sampled from a standard normal distribution and $\delta$ was sampled from an inverse gamma distribution (shape parameter 5, scale parameter 2), following the original work by Johnson et al.[25].

Goh and Wong published the *D2.2* series of 100 simulated proteomic datasets with two batches and simulated conditions, which we used for additional validation of the BERT algorithm[37]. Raw data was log-transformed, and zeros were masked as NA. Proteins with zero variance in any of the batches were removed, and ASW scores of the raw and batch-effect corrected data were computed. Differential expression (DE) analyses were performed by $t$ tests with Benjamini-Hochberg correction, and $p$-values $< 0.05$ were considered significant (only complete data with variance greater than $10^{-5}$). DE results were quantified *per protein* using the $F_1$-score w.r.t the simulated, true significance.

### *omic* Data
Petralia et al. published an integrated proteogenomic analysis of 218 pediatric brain tumors from 7 entities[39] (processed data here). The authors used MS3 and liquid chromatography and processed the data in 23 TMT batches. We processed the raw data using MaxQuant[50] and identified 9155 proteins. The authors defined 8 proteomic tumor groups, which we used as biological labels. Raw intensities were log-transformed and median-normalized.

Krug et al. published a dataset of 125 treatment naïve breast tumors that were characterized using LC-MS/MS with tandem mass tags using a benchtop Orbitrap Fusion Lumos mass spectrometer[38] (raw and processed data available). We obtained the processed data directly from the authors. 13769 proteins were quantified in total. We used the PAM50 subtypes as biologic labels for the ASW computation. We excluded the "normal" subtype from our considerations, because only 5 samples of this type were measured. For three further samples, PAM50 annotations were unavailable. Raw intensities were log-transformed and median-normalized.

Nusinow et al. published a collection of proteomic measurements of 375 cell lines for the Cancer Cell Line Encyclopedia[40] (processed data available). The authors used HPLC and MS3 to process 378 unique samples in 42 TMT batches and quantified a total of 12970 proteins. The tissue of origin was used as biologic label for ASW computation. Raw intensities were log-transformed and median-normalized.

For their characterization of HarmonizR, Voss and Schlumbohm et al. published a dataset of four batches, which comprise a total of 25 samples[24] (mouse models for SHH-medulloblastoma and healthy controls, PRIDE accession number PXD027467). Half of the samples were prepared using formalin fixation and paraffin-embedding (FFPE), whereas the other half was snap frozen (fresh frozen, FF). DDA-LFQ was used for protein quantification. We considered the biological condition of the mice as the label and excluded batch 1 due to insufficient data per label ($n = 8$ samples). Raw intensities were log-transformed and median-normalized.

We used three publicly available datasets of gene expression profiling with the GeneChip technology of Affymetrix and different sample processing protocols[51,52] (GEO accession numbers GSE18520, GSE66957, GSE69428). We considered high-grade serous ovarian carcinoma, as well as normal controls and excluded samples of type FTSC from GSE69428. No pre-processing steps were performed for this work.

Wehrens et al. published three datasets for the metabolomic characterization of *Arabidopsis Thaliana*, and data generation and pre-processing details can be found in the original publication[43] (processed data obtained from GitHub). For the first dataset, the authors used LS-MS with an LTQ-Orbitrap hybrid MS system to quantify 567 metabolites across 10 batches of 48–80 samples (761 samples in total). For the second dataset, the authors used GC-MS with Agilent GC7890A and a quadrupole MSD Agilent 5978C to quantify 603 volatile organic compounds across 15 batches of 34–99 samples (753 samples in total). The third dataset (polar extracts from a nucleotype-plasmotype combination study) was created using GC-ToF-MS (Agilent 6890 GC coupled to a Leco Pegasus III MS), quantifying 75 metabolites across 4 batches of 31–89 samples (240 samples in total).

The ADHD-200 consortium established a large-scale, multi-center dataset of fMRI dataset with accompanying clinical (phenotypical) information[53]. Later, the neuro bureau published preprocessed phenotypic information via the Neuroimaging Informatics Tools and Resources Clearinghouse (NITRC)[44]. In total, 300 of these preprocessed samples were selected from the Peking University (245 patients), the Kennedy Krieger Institute (94 patients) and the New York University Langone Medical Center (261 patients). Considerations

were restricted to the Attention-Deficit Hyperactivity Disorder (ADHD) index, the inattentiveness, the hyper/impulsiveness, the verbal IQ, the performance IQ and the full 4 IQ. No further preprocessing was performed. Further details can be found in the Supplementary Information.

### SVC-Based classification
The *Krug* dataset was selected to investigate the efficacy of BERT for downstream machine learning tasks. Support-vector classifiers (SVCs) were trained in 10 independent repetitions, leaving one batch out as a test set at a time (leave-one-batch-out cross-validation). Per repetition and left-out batch, 50 iterations of randomized grid-search were used to maximize the balanced accuracy of the following procedure in 10-fold cross-validation across the remaining data: Features with at most 10% missing values were selected and restricted to the 5% subset with the highest variance. Min-max scaling or standardization (the choice of which was optimized as part of the hyperparameter optimization) was employed, followed by k-Nearest-Neighbor imputation (between 1 and 5 neighbors and uniform/distance-based weights, both optimized as hyperparameters). Finally, an SVC was trained using an optimized hyperparameter combination (RBF/linear kernel, regular or balanced class weights, regularization $C \in [10^{-3}, 10^2]$, all other parameters left at their default in scikit-learn[54]). Finally, balanced test accuracy was measured on the left-out test set. Here, imputation was used to allow this simple demonstration, but imputation-free methods can be used as well[55–59].

Note that kNN-imputation was selected due to its computational efficiency, which allows for the excessive hyperparameter optimization performed here. Iterative methods, such as the popular *missForest* algorithm, have been recommended[60,61] but are computationally too expensive on datasets of this size. Proof-of-principle experiments with a single hyperparameter combination of an iterative imputer (*scikit-learn* library, Bayesian Ridge regressor) and with the efficient *missMDA*[62–65] method confirm the validity of the obtained results ($p$-value of $p < 9 \cdot 10^{-10}$ from one-tailed $t$ tests – Cohen's $d \approx 397$ and p-value of $p < 7 \cdot 10^{-13}$ from one-tailed $t$ tests – Cohen's $d \approx 319$, respectively).

### Hardware and dependencies
All experiments have been conducted on nodes of the cluster *Maxwell* at DESY. If not explicitly specified otherwise, the experiments in this work have been conducted using the latest versions of BERT and HarmonizR.

For our experiments, we used R version 4.4.2 (2024-10-31) with BERT 1.3 and HarmonizR 1.6. For data analysis and classification experiments, we used Python 3.12.10 with *scikit-learn* 1.6.1, *matplotlib* 3.10.1, *seaborn* 0.13.2, *pandas* 2.2.3 and *numpy* 2.2.4. We provide a reproducible environment via a container definition file (*apptainer*) in the GitHub repository for experiments (https://gitlab.desy.de/yannis.schumann/bert_experiments).

### Reporting summary
Further information on research design is available in the Nature Portfolio Reporting Summary linked to this article.

## Data availability
Source data are provided with this paper. Study results are based on third-party data, which is publicly available as detailed in the respective original publications. The proteomic data by Petralia et al.[39] used in this study are available in the Proteomics Data Commons database under accession code PDC000180. The proteomic data by Krug et al.[38] used in this study are available in the Proteomics Data Commons database under accession code PDC000120. The proteomic data by Nusinow et al.[40] used in this study are available in the MassIVE database under accession code MSV000085836 [https://massive.ucsd.edu/ProteoSAFe/

dataset.jsp?task=02cd1b6a7c674f3ebdbed300b5d9aa57]. The proteomic data by Voss and Schlumbohm et al.[24] used in this study are available in the PRIDE database under accession code PXD027467. The simulated proteomic data by Goh and Wong[37] are available directly from their publication [https://doi.org/10.1186/s12864-017-3490-3]. The transcriptomics data used in this study are available in the GEO database under accession codes GSE18520, GSE66957 and GSE69428. The metabolomic data by Wehrens et al.[43] used in this study are available via GitHub [https://github.com/rwehrens/BatchCorrMetabolomics/tree/master/data]. Any additional data processing is described in Section Methods. We further publish an extended repository with all scripts for simulations, *omic* data analyses and publication-ready plots, where all elements of the study are encapsulated into appropriate *make* targets. Furthermore, we also provide the definition file for a container in which we executed all our experiments to allow for maximum reproducibility. The BERT software is publicly available, as reported in Section Code Availability. Source data are provided in this paper.

## Code availability
The BERT algorithm is publicly available and has been deposited in GitHub at https://github.com/HSU-HPC/BERT, under the GPL-3 license. It is further provided as an R package via Bioconductor. The specific version of the code associated with this publication is archived in Zenodo and is accessible via Zenodo[66].

The documentation includes extensive installation instructions and working examples. User support is provided via the Bioconductor forum and the GitHub issue section.

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

## Acknowledgements
Computational resources (HPC-cluster HSUper) have in part been provided by the project hpc.bw, funded by dtec.bw - Digitalization and Technology Research Center of the Bundeswehr. dtec.bw is funded by the European Union - NextGenerationEU. This research was supported in part through the Maxwell computational resources operated at Deutsches Elektronen-Synchrotron DESY, Hamburg, Germany. J.E.N. is funded by the DFG (Emmy Noether program).

## Author contributions
Y.S. and S.S. conceived the core BERT algorithm, and Y.S. implemented it in R and extended the functionality. S.S. implemented the previously published HarmonizR algorithm. Y.S. conducted the experiments, analyzed the data and wrote the manuscript. J.E.N. and P.N. supervised the study. All authors have read and approved the final manuscript.

## Funding

## Competing interests
The authors declare no competing interests.
