## [Transparent Peer Review file · Nature Communications]

High Performance Data Integration for Large-Scale Analyses of Incomplete Omic Profiles Using Batch-Effect Reduction Trees (BERT)

Corresponding Author: Dr Yannis Schumann

Version 0:

Reviewer comments:

Reviewer #1

(Remarks to the Author)

The authors have developed an algorithm BERT to integrate omic data sets after batch correction. They have carefully provided responses to previous referees' comments. After reading all responses and the main manuscript, I observe that while this method might be a good candidate for batch correction in multiple data sets, some more clarifications are required.

1. Pairs of batches are considered at each tree level. If this is done by randomly selecting each pair, some justification is needed about the robustness of the result i.e. the result is independent of the pairs chosen in such manner. This needs to be established theoretically or mathematically rather than using simulations.
2. Line 74 – 76: There is some confusion in understanding at this part. It seems that the problem discussed in the manuscript is how to correct batch effects for omic data, for example, transcriptome data from different batches. What is meant by “All other features, for which the numerical values exclusively originate from either one or the other input batch (due to pre-processing the other batch contains only missing values), are per definition not afflicted by batch-effect”? If one data set does not contain a particular feature that exists in another data set, how can we ensure that it is free from batch-effect? May be, we are unable to say; but it is possible that it still has some batch effects. However, this probably happens because we are looking at a pair of data batches. Does the same issue still exist if a different data set is chosen as the pair of the first one? Some more insight at this issue is required for more clarification.
3. Although the authors explained in response to reviewers' comments that choice of P, R and S does not affect the output quality, some questions remain. After the parallelization stops at a particular S, the intermediate batches are integrated sequentially. Whenever an algorithm stops based on a stopping criterion in a sequential modality, the result should depend on that sequential procedure and the stopping rule. So, it is expected that it might get affected by the choice of S. Some more insight, preferably not through simulation, would strengthen this claim.
4. Line 100 – 104: It seems that BERT is already in Bioconductor after peer reviewing. To publish in Nat Comm journal, more mathematical/statistical/theoretical justification should be there. Nowhere, any property has been evaluated; everything is justified by observations through simulated data sets. This establishes its usage weakly but cannot ensure the strength of this method in broader perspective.
5. Line 254 – 255: This statement is slightly confusing. Does it mean multiple data sets generated in different batches for one type of omic data, for example, transcriptome data, or BERT can correct data sets from different types of omic data in a single set up? This needs to be clearly stated and explained in view of the main objective of the paper. Features in different omic data might be different; even if they are somewhat same, it is not clear to what extent there is an overlap of sets of features.

This manuscript may be considered after revision. Decision would be taken after the revision.

(Remarks on code availability)

Report on the manuscript High Performance Data Integration for Large-Scale Analyses of Incomplete Omic Profiles Using Batch-Effect Reduction Trees (BERT) (NCOMMS-25-09274-T) by Schumann et al.

The authors have developed an algorithm BERT to integrate omic data sets after batch correction. They have carefully provided responses to previous referees' comments. After reading all responses and the main manuscript, I observe that while this method might be a good candidate for batch correction in multiple data sets, some more clarifications are required.

1. Pairs of batches are considered at each tree level. If this is done by randomly selecting each pair, some justification is needed about the robustness of the result i.e. the result is independent of the pairs chosen in such manner. This needs to be established theoretically or mathematically rather than using simulations.
2. Line 74 – 76: There is some confusion in understanding at this part. It seems that the problem discussed in the manuscript is how to correct batch effects for omic data, for example, transcriptome data from different batches. What is meant by “All other features, for which the numerical values exclusively originate from either one or the other input batch (due to pre-processing the other batch contains only missing values), are per definition not afflicted by batch-effect”? If one data set does not contain a particular feature that exists in another data set, how can we ensure that it is free from batch-effect? May be, we are unable to say; but it is possible that it still has some batch effects. However, this probably happens because we are looking at a pair of data batches. Does the same issue still exist if a different data set is chosen as the pair of the first one? Some more insight at this issue is required for more clarification.
3. Although the authors explained in response to reviewers' comments that choice of P, R and S does not affect the output quality, some questions remain. After the parallelization stops at a particular S, the intermediate batches are integrated sequentially. Whenever an algorithm stops based on a stopping criterion in a sequential modality, the result should depend on that sequential procedure and the stopping rule. So, it is expected that it might get affected by the choice of S. Some more insight, preferably not through simulation, would strengthen this claim.
4. Line 100 – 104: It seems that BERT is already in Bioconductor after peer reviewing. To publish in Nat Comm journal, more mathematical/statistical/theoretical justification should be there. Nowhere, any property has been evaluated; everything is justified by observations through simulated data sets. This establishes its usage weakly but cannot ensure the strength of this method in broader perspective.
5. Line 254 – 255: This statement is slightly confusing. Does it mean multiple data sets generated in different batches for one type of omic data, for example, transcriptome data, or BERT can correct data sets from different types of omic data in a single set up? This needs to be clearly stated and explained in view of the main objective of the paper. Features in different omic data might be different; even if they are somewhat same, it is not clear to what extent there is an overlap of sets of features.

Reviewer #2

(Remarks to the Author)

Schumann et al. provided a significantly improved version of their manuscript titled 'High Performance Data Integration for Large-Scale Analyses of Incomplete Omic Profiles Using Batch-Effect Reduction Trees (BERT).'

This review refers only to items that were answered in an unsatisfactory manner. All other points were addressed appropriately by the authors.

Minor comments

> Originally, we had tried to use the missForest-method described by Stekhoven et al. (DOI: 10.1093/bioinformatics/btr597), since it was positively evaluated in literature (e.g., Jin et al., DOI: 10.1038/s41598-021-81279-4).

It is difficult to agree that missForest would be the appropriate imputation method in this setting. Even though missForest performs well in the benchmarks, its time of execution does not scale well enough for larger datasets. In such a situation, it might be better to consider missMDA (10.18637/jss.v070.i01), which both performs well and is very fast even in large benchmarks (10.1093/bioinformatics/btae098).

> Our results show close-to identical behavior of BERT for up to 50% missing values of MCAR and MNAR type, from which we conclude that our focus on the original data-generating process is justified for the main manuscript. Of note, BERT and HarmonizR showed equal performance for all missing value mechanisms, although they rely on strongly different schemes (binary tree, divide-and-conquer scheme).

The simulation is greatly improved. Was it implemented in https://gitlab.desy.de/yannis.schumann/bert_experiments/-/blob/main/Simulation/MCARvsMNAR/MVTypes.R?ref_type=heads#L11 ? In this case, does Figure 2 B.2 present results for both MNAR and MCAR scenarios? Please clarify.

(Remarks on code availability)

The code is reproducible and passes demanding (if compared to CRAN) Bioconductor checks. It is a strong point of BERT from the get-go.

Reviewer #4

(Remarks to the Author)

The authors addressed all major comments done before and it is ready to be accepted.

(Remarks on code availability)

Version 1:

Reviewer comments:

Reviewer #1

(Remarks to the Author)

The authors responded all queries. The manuscript may be accepted.

(Remarks on code availability)

Looks okay.

Reviewer #2

(Remarks to the Author)

I accept the changes and congratulate the authors on a well-executed publication. I would inquire about the link to the repository because I could not see it in the manuscript: https://gitlab.desy.de/yannis.schumann/bert_experiments/

(Remarks on code availability)

The code is reproducible and passes Bioconductor checks. It installs on my machine and runs the example code.

Reviewer 1

The authors have developed an algorithm BERT to integrate omic data sets after batch correction. They have carefully provided responses to previous referees' comments. After reading all responses and the main manuscript, I observe that while this method might be a good candidate for batch correction in multiple data sets, some more clarifications are required.

We thank the reviewer for the friendly and constructive summary and describe below how we accommodated for each of their specific comments.

1. Pairs of batches are considered at each tree level. If this is done by randomly selecting each pair, some justification is needed about the robustness of the result i.e. the result is independent of the pairs chosen in such manner. This needs to be established theoretically or mathematically rather than using simulations.

We highly appreciate the reviewers interest and significantly expanded our submission to provide the requested theoretical formalism (Suppl. Information Sec. 1, pp. 2-8). For readability of this response, we additionally summarize the central points of our argument in the following:

BERT hierarchically applies the established algorithms ComBat or limma on a binary tree of pairwise batch-effect correction steps. The processing order of batches (i.e., the selection of pairs) is determined by their order in the user input and the BERT output is hence reproducible and not subject to any random procedure. In our novel section 1 of the supplementary information, we show by straightforward computation on a four-batch subtree that the hierarchical data integration yields equivalent results as integration in a single data integration step. We further demonstrate via symbolic calculation that this behavior generalizes to variable number of samples per batch.

Moreover, the robustness of the BERT output with respect to batch input order (i.e., the selected pairs) can be established theoretically for limma, in addition to the already provided experimental validation (Supplementary Information Sec. 1.1.4 ll. 137-154). Note that this strict guarantee holds for $N=2^m$ batches with complete data where $m \in \mathbb{N}$. For all other cases, a permutation of batches may introduce an additive offset on each feature (constant across all batches), cf. Sec. 1.2.3 of the Supplementary Information for more details (ll. 85-103). We argue however, that this offset is irrelevant for the anticipated workflows and downstream tasks of users including differential expression analysis, classification and others, which exclusively rely on *relative differences* of expression values between samples. Moreover, users will typically apply domain-specific normalization procedures to each feature after batch-effect correction, such as z-score normalization, which diminishes such introduced offset.

For the ComBat method, the corrected data is computed using parameters from robust, constrained least-squares optimization and an estimate of the biological/technical variation from Bayesian statistics. While a similar analysis of robustness would be possible for the feature-wise least-squares parameters, such a statement can not be established easily for the Bayes parameters due to the heuristic stopping criterion of ComBat's parameter estimation method.

However, we conduct extensive experimental error analyses for ComBat that show convincingly that these parameters also do not lead to any instability (Suppl. Inf. Sec. 1.1.3-1.1.4, ll. 105-135).

We have extended the Supplementary Information by a novel section 1 describing the mathematical formalism of limma and ComBat as well as a justification of the hierarchical BERT approach. Please note that we have collated the theoretical analysis with the respective experimental validation (incl. the analysis of error propagation across tree levels) as they are strongly related. We have further extended the results section of the main manuscript to highlight the robustness of BERT output and the preservation of relative expression differences independent of batch order and point out potential limitations to users (ll. 147-154).

2. Line 74 – 76: There is some confusion in understanding at this part. It seems that the problem discussed in the manuscript is how to correct batch effects for omic data, for example, transcriptome data from different batches. What is meant by “All other features, for which the numerical values exclusively originate from either one or the other input batch (due to pre-processing the other batch contains only missing values), are per definition not afflicted by batch-effect”? If one data set does not contain a particular feature that exists in another data set, how can we ensure that it is free from batch-effect? May be, we are unable to say; but it is possible that it still has some batch effects. [continued below]

We highly appreciate the comment and agree with the reviewer that this statement was imprecise. Given a pair of batches where a subset of features has only been quantified in one of the batches, the respective numerical values may indeed contain batch-specific biases. However, such bias (i.e., batch-effect) can not be quantified or corrected, since no *relative* information is known from the other batch on that tree level. Therefore, BERT propagates these values to the next tree level until they will be corrected for batch-effects using another batch with suitable numerical data for the respective features. Note that neither limma nor ComBat would be able to consider these features naively and this capability represents a unique feature of BERT. We have rephrased the referenced statement as “[these values]...are propagated to the next tree level without further changes” throughout the entire manuscript to make this more clear (e.g., p. 5, ll. 76 and Fig. 2 A).

2. [continued] However, this probably happens because we are looking at a pair of data batches. Does the same issue still exist if a different data set is chosen as the pair of the first one? Some more insight at this issue is required for more clarification.

We appreciate the interest of the reviewer. Indeed, the order of batches (i.e., the selected pairs) does not influence the final, standardized values for these features. This is discussed in the novel, theoretical justification of the hierarchical BERT approach in Sec. 1 of the Supplementary Information, which has been added to accommodate for comment 1 of this reviewer.

3. Although the authors explained in response to reviewers’ comments that choice of P, R and S does not affect the output quality, some questions remain. After the parallelization stops at a particular S, the intermediate batches are integrated sequentially. Whenever an algorithm stops based on a stopping criterion in a sequential modality, the result should depend on that sequential procedure and the stopping rule. So, it is expected that it might get affected by the choice of S. Some more insight, preferably not through simulation, would strengthen this claim.

We highly appreciate the interest of the reviewer and are happy to elaborate on this. Indeed, the parameter S represents the number of intermediate batches at which parallelization stops and the remaining batches are integrated sequentially (cf. Fig. 2 A of this response). S is specified directly by the user when calling the BERT routine.

Due to the tree structure of the algorithm, integration steps from different branches are strictly independent. Therefore, they may be executed in parallel or sequentially in any given order without any impact on the result (cf. Fig. 2 B of this response). Thus, the BERT output does not depend on the value of S . We agree with the reviewer that this is a specific property of the considered tree structure and should have been made more clear in the manuscript. We have incorporated a corresponding statement into the Supplementary Information, p.10, ll. 156-158.

Figure 2 Concept sketch of BERT parallelization scheme and the parameter S . **A** BERT tree with 16 input batches, indicating the stop of parallelization for $S=2$ and $S=4$. **B** Order of batch-effect correction steps for two processes (left, processes A and B) and for fully sequential execution (right). Results are strictly identical.

4. Line 100 – 104: It seems that BERT is already in Bioconductor after peer reviewing. To publish in Nat Comm journal, more mathematical/statistical/theoretical justification should be there. Nowhere, any property has been evaluated; everything is justified by observations through simulated data sets. This establishes its usage weakly but cannot ensure the strength of this method in broader perspective.

We thank the reviewer for raising this important concern. We agree with the reviewer that our submitted manuscript focuses on a phenomenological characterization of BERT instead of mathematical formalism, however far beyond mere simulations. We had deliberately chosen this style of writing to address the particularly broad readership of Nature Communications (cf. aims & scope of the journal), which includes researchers from various fields of applications (e.g., biology, health, social sciences), for whom the applicability and thorough characterization will be most relevant.

However, we underwent significant efforts to add additional mathematical/theoretical formalism to the manuscript by

- detailing the mathematical principles of ComBat and limma, which are fundamental for BERT (Sec. 1.1.1-1.1.2 of Supplementary Information)
- providing mathematical proof for the validity of the hierarchical BERT approach compared to a classical single-step correction (Sec. 1.1.3 of Supplementary Information)
- providing an in-depth analysis of BERT robustness to permutations of the input batches (i.e., the selected pairs of batches, cf. comment 1 of this reviewer, Sec. 1.1.3 and Sec. 1.1.4 of Supplementary Information)
- and analyzing the algorithmic complexity of BERT (Sec. 6 of Supplementary Information).

Providing proof of BERT's capabilities beyond the scope of simulated data is of major concern to us. We therefore showed significant evidence and characterization on *real* (i.e. not simulated) data throughout the submitted manuscript and made sure to utilize a variety of real-world metrics to capture different properties of the method, including

- runtime and scaling (e.g., Fig. 4 panels B.2, B.3, D.1; Fig. 5 F.1)
- ASW scores for both batch and label, as well as retained numerical values (e.g., Fig. 4 panels E.1, F.1; Tab. 1; Fig. 5 D.1, D.2)
- dimensionality-reduction techniques and classification (e.g., Fig. 4 C.1-C.3, Fig. 5 A.1-C.2)

on a rich pool of representative datasets from proteomics, transcriptomics, metabolomics and beyond. Furthermore, our implementation of BERT provides automated user-feedback of ASW scores, execution time and retained numerical values into the BERT algorithm, which will immediately inform users about the strength of the method for their application.

We firmly believe that the revised manuscript shows extensive justification of the proposed method, including mathematical formalism and experimental proof on *both* simulated and real data.

5. Line 254 – 255: This statement is slightly confusing. Does it mean multiple data sets generated in different batches for one type of omic data, for example, transcriptome data, or BERT can correct data sets from different types of omic data in a single set up? This needs to be clearly stated and explained in view of the main objective of the paper. Features in different omic data might be different; even if they are somewhat same, it is not clear to what extent there is an overlap of sets of features.

We thank the reviewer for this important remark. BERT does indeed consider only one, single-omic dataset at a time, potentially split in multiple batches. We agree with the reviewer that the sentence in the original manuscript was slightly confusing and have rephrased the statement accordingly (p. 12, l. 261).

Remarks on code availability: ...[same comments as remarks to authors]

The reviewer's remarks on code availability are identical to the remarks detailed above. We therefore provide the answers only once.

Reviewer 2

Schumann et al. provided a significantly improved version of their manuscript titled 'High Performance Data Integration for Large-Scale Analyses of Incomplete Omic Profiles Using Batch-Effect Reduction Trees (BERT).'

This review refers only to items that were answered in an unsatisfactory manner. All other points were addressed appropriately by the authors.

We thank the reviewer for the positive and constructive feedback and describe below how we accommodated for each of his/her specific comments.

1. It is difficult to agree that missForest would be the appropriate imputation method in this setting. Even though missForest performs well in the benchmarks, its time of execution does not scale well enough for larger datasets. In such a situation, it might be better to consider missMDA (10.18637/jss.v070.i01), which both performs well and is very fast even in large benchmarks (10.1093/bioinformatics/btae098).

We highly appreciate this recommendation by the reviewer. We have conducted additional experiments with missMDA imputation of both the raw and imputed dataset and found the results to agree with our prior experiments using iterative and KNN-based imputation. We have expanded our manuscript by these additional results and included the suggested references (p. 17, ll. 410-415)

2. The simulation is greatly improved. Was it implemented in https://gitlab.desy.de/yannis.schumann/bert_experiments/-/blob/main/Simulation/MCARvsMNAR/MVTypes.R?ref_type=heads#L11 ? In this case, does Figure 2 B.2 presents results for both MNAR and MCAR scenarios? Please clarify.

We thank the reviewer for the positive feedback and the question. Since Figure 2 does not present a panel B.2 we respectfully assume that a minor typo has happened here and the reviewer was referring to Figure 3 B.2 of the main manuscript instead, which shows the ASW scores of BERT output for variable amount of missing values.

A joint plot with the results from MNAR and MCAR scenarios is shown in Fig. 4 A.1 on p. 11 of the Supplementary Information, for which the simulations were indeed implemented with the referenced source file. Since these additional simulations had verified that the behavior is very similar between MCAR and MNAR scenarios, the referenced panel in the main manuscript only includes results for the MCAR scenario to maintain consistency with the other panels. We have included a reference to the comparison with MCAR data in the main manuscript to make this more clear to the reader (main manuscript p. 7, l. 137).

Please note that a typo was removed from the legend in supplementary figure Fig. 4 A.1 (formerly Fig. 3 A.1), where the MNAR results were accidentally labeled as MCAR. We sincerely apologize for any confusion this may have caused.

Remarks on code availability: The code is reproducible and passes demanding (if compared to CRAN) Bioconductor checks. It is a strong point of BERT from the get-go.

We thank the reviewer for this positive feedback.

Reviewer 4

Summary: The authors addressed all major comments done before and it is ready to be accepted.

We thank the reviewer for this positive feedback.

Reviewer 1

The authors responded all queries. The manuscript may be accepted.

We thank the reviewer for the friendly and constructive review throughout this review phase of the manuscript.

Remarks on code availability: Looks okay

We thank the reviewer for their expert opinion.

Reviewer 2

I accept the changes and congratulate the authors on a well-executed publication. I would inquire about the link to the repository because I could not see it in the manuscript: https://gitlab.desy.de/yannis.schumann/bert_experiments/

We thank the reviewer for the positive and constructive feedback throughout the review phase. The link to the repository can be found in line 448 of the main manuscript (left-click on the term “extended repository”) and in ll. 428.

Remarks on code availability: The code is reproducible and passes Bioconductor checks. It installs on my machine and runs the example code.

We highly appreciate this appraisal by the reviewer.